# Feeding Thai Native Sheep Molasses Either Alone or in Combination with Urea-Fermented Sugarcane Bagasse: The Effects on Nutrient Digestibility, Rumen Fermentation, and Hematological Parameters

**DOI:** 10.3390/vetsci9080415

**Published:** 2022-08-06

**Authors:** Thaintip Kraiprom, Sitthisak Jantarat, Suphawadee Yaemkong, Anusorn Cherdthong, Tossaporn Incharoen

**Affiliations:** 1Faculty of Science and Technology, Pattani Campus, Prince of Songkla University, Pattani 94000, Thailand; 2Faculty of Food and Agricultural Technology, Pibulsongkram Rajabhat University, Phitsanulok 65000, Thailand; 3Increase Production Efficiency and Meat Quality of Native Beef and Buffalo Research Group, Department of Animal Science, Faculty of Agriculture, Khon Kaen University, Khon Kaen 40002, Thailand; 4Department of Agricultural Science, Faculty of Agriculture Natural Resources and Environment, Naresuan University, Phitsanulok 65000, Thailand

**Keywords:** alternative roughage sources, molasses, nutrient digestibility, ruminant, sugarcane bagasse, sheep, urea treatment, volatile fatty acid

## Abstract

**Simple Summary:**

A key strategy for attaining value-added and ecologically sustainable operations is the conversion of agro-industrial waste into animal feed. Sugarcane bagasse, a fibrous byproduct of the sugar-refining process, is widely available. Bagasse, however, has low nutritional value and a high quantity of indigestible fiber, which causes low digestibility and, as a result, poor animal performance. Ensiled bagasse’s fermentation properties improved after being treated with molasses and urea. Sheep’s ability to digest crude protein was increased by mixing fermented sugarcane bagasse with 10% molasses and 3% urea without having an adverse impact on the ruminal fermentation or the animal’s hematological parameters.

**Abstract:**

The purpose of this study was to find out how adding molasses to fermented sugarcane bagasse (FSB) alone or in combination with urea affected sheep’s rumen fermentation, hematological parameters, and ability to digest nutrients. Four Thai native sheep with an initial body weight (BW) of 20.87 ± 1.95 kg and 11 ± 1.0 months old were assigned to a 4 × 4 Latin square design with 4 periods of 14-d adaptation and 7 d of sample collection. Each treatment received a different combination of experimental roughage as follows: FSB without additives (T1), FSB + 10% molasses (T2), FSB + 20% molasses (T3), and FSB + 10% molasses + 3% urea (T4). The concentrate diet was fed twice daily at 2% BW, while roughage sources were provided ad libitum for each treatment. The crude protein (CP) digestibility in the T2 and T3 groups was higher (*p* < 0.05) than in the FSB group without additions, with the T4 group having the highest (*p* < 0.05). Although there were no significant differences in blood glucose, packed cell volume, ruminal pH, ammonia–nitrogen (NH_3_-N), propionic acid, or acetic acid, the plasma urea nitrogen (PUN) at 0 h was highest in the T4 group (*p* < 0.05) compared with the other groups. However, the proportion of butyric acid tended to be higher in all FSB groups with additives. Thus, the current experiment concluded that the addition of molasses alone or in combination with urea had positive effects on pH and LAB population, and including both together in FSB improved the CP digestibility of sheep. In conclusion, FSB with 10% molasses and 3% urea might be used as an alternate roughage source for ruminants without affecting the animal’s ruminal fermentation or hematological parameters.

## 1. Introduction

Small ruminants play a crucial role in the food security and nutrition of millions of inland people, especially in undeveloped and developing countries [1]. These animals can potentially convert low-quality feed resources into high-quality protein sources, alleviating malnutrition and contributing to a sustainable food system [2]. Sheep are one of the most significant ruminant species, and they are generally maintained for wool, sheepskin, meat, and milk production. In Thailand, the sheep population continues to grow and is mostly raised in small farming villages in the southern region. However, the key to the success of sheep producers seems to be both the availability and quality of roughage sources, especially during the dry season [3]. Many sheep are maintained on poor-quality fibrous feeds, depending on the availability of crop residues, agro-industrial byproducts, and nonconventional feeds.

Sugarcane is one of the most important economic crops globally and is mostly used as a raw material for the sugar-refining industry. Approximately 80% of the sugar produced from sugarcane is grown in tropical and subtropical climates, with approximately 27 million ha in total [4]; worldwide sugar production was approximately 166.18 million metric tons in the 2019 to 2020 growing season [5]. The primary waste product in the sugar refining industry is bagasse, which remains after the juice is extracted from the cane stem. Sugarcane bagasse is the fibrous residue of this process and is available in enormous quantities; thus, it could be used as an alternative roughage source for ruminant feed [6]. This concept might be an important means of achieving value-added and environmentally friendly operations, due to the conversion of agro-industrial waste into animal feed. However, So et al. [7] noted that sugarcane bagasse has low nutritional value and high indigestible fiber content as dry matter (DM), such as protein (2.67%), ether extract (0.31%), cellulose 54.61%), hemicellulose (5.51%), lignin (14.29%), and gross energy (6.93 kcal/g), resulting in low digestibility (26.7%) and, consequently, poor animal performance [6]. Therefore, animal nutritionists have studied several techniques aimed at enhancing the nutritional content and utilization of sugarcane bagasse. The increase in dry matter intake as a result of molasses’ improved palatability has already been recognized as one of the consequences of including it in ruminant diets [8]. In the case of grazing ruminants fed with subpar fodder, this outcome is much more significant. Under these circumstances, molasses has a stimulating impact on the ruminal microbiota’s digestive activity, which enhances the digestion of coarse grade feed and the intake of dry matter [7]. Wang et al. [9] found that adding molasses at levels of 2.5 to 5.0% to mixed roughage increased the lactic acid content and decreased the pH and NH_3_–N contents, resulting in improved silage quality. Roughage treated with 7.5% molasses as an additive had better DM degradation and aerobic stability [10]. Adding molasses at a level of 4% to an entire wheat crop improved the ensiling characteristics and quality, leading to well-preserved silage as well as a reduction in nutrient loss [11]. So et al. [12] reported that using sugarcane bagasse treated with molasses, along with *Lactobacillus* and cellulase for use as a ruminant feed, enhanced the overall productivity and ruminal fermentation as well as improved nitrogen (N) and energy utilization. In addition, treating ensiled roughage with urea is an approved method of improving the nutritional value and quality of silage products. Gunun et al. [13] reported that urea-treated silage ameliorated feed intake, nutrient digestibility, and ruminal fermentation, resulting in increased microbial N synthesis in dairy steers. There is a wide range in the amount of protein consumed. A normal Western diet contains between 70 and 100 g of protein per day, with about 50 to 60 g coming from endogenous sources such as gastrointestinal secretions [14]. Microbial protein synthesis is one indicator that rumen microorganisms are producing proteins for use as ruminant protein sources. Because microbial protein produced in the rumen provides half of the amino acids needed by ruminants, microbial protein synthesis is significant in ruminants. An adequate supply of energy (ATP) from the rumen’s fermentation of organic matter and nitrogen (N) from the breakdown of sources of both non-protein and protein nitrogen is essential for the synthesis of microbial protein and the growth of ruminal microbes [15]. These requirements can be influenced by environmental or dietary factors. This suggests that urea treatment could increase the crude protein (CP), leading to higher NH_3_–N retention in sugarcane bagasse fermentation [6]. An ammoniation procedure can be performed to conserve a high moisture level in fresh roughage [16] and improve the nutrient profile of silage by increasing the non-protein nitrogen (NPN) source and degrading the fibrous fraction [17]. For large ruminant animals’ diets, the idea of adding molasses and urea to improve fermentation quality sugarcane bagasse has been widely applied. Lunsin et al. [18] described that the CP content and in vitro DM and organic matter (OM) digestibility of dairy cows were significantly (*p* < 0.05) increased in sugarcane bagasse treated with a combination of 5% urea and 5% molasses. Similarly, the inclusion of 2% urea + 2% Ca(OH)_2_ as additives in sugarcane bagasse fermentation improved feed intake, digestibility, and rumen fermentation in beef cattle [6]. However, there is little information on feeding sheep fermented sugarcane bagasse with additives, especially during the dry season when sheep have less access to green pasture. In order to improve feed utilization in sheep, we hypothesized that sugarcane bagasse may be enhanced by being fermented with molasses or urea. To this end, the present study was performed to prepare the fermented sugarcane bagasse (FSB) with molasses alone and in combination with urea to assess their effects on the nutrient digestibility, hematological parameters, and ruminal fermentation of Thai native sheep.

## 2. Materials and Methods

### 2.1. Preparation of Fermented Sugarcane Bagasse

Bagasse was obtained from a sugarcane juice producer in Pattani Province, Thailand. The bagasse was chopped into 2.0–3.0 cm pieces using an electric cutting machine and temporarily kept in a plastic box. Fermented sugarcane bagasse (FSB) was prepared by adding molasses at levels of 0, 10, and 20% in respective group, while another sample of bagasse was merged with a mixture of 10% molasses and 3% urea, according to a ratio of 100 L of water to 100 kg sugarcane bagasse (air-dry basis). Each treated sample of bagasse was thoroughly sprayed with the prepared solution, preserved in a polyethylene tank, and covered with a vinyl sheet for 30 d before use.

### 2.2. Animals and Experimental Design

Ethical procedures and animal care practices were regulated and approved by the Institutional Animal Care and Use Committee, Prince of Songkla University (Ref. 40/2018), Thailand, and the trial was performed according to the Ethical Principles and Guidelines for the Use of Animals for Scientific Purposes, National Research Council of Thailand. The sheep used in this study were obtained from Prince of Songkla University’s Animal Farm, Faculty of Science and Technology. Four crossbred male sheep (Thai native x Kelantan) aged 11 ± 1.0 months old (20.87 ± 1.95 kg of initial BW) were randomly assigned to four groups according to a 4 × 4 Latin square design. Each of the four sheep received each of the four treatment regimens for 21 days before rotating to another one of the same four treatment regimes. As a result, every individual sheep received each of the four treatment regimes. This enables n = 4 replicate animals (as well as samples) for each treatment. Each group received different experimental roughage as follows: FSB without additives (T1), FSB + 10% molasses (T2), FSB + 20% molasses (T3), and FSB + 10% molasses + 3% urea (T4). The concentrate diet was formulated to meet requirements for maintenance according to NRC [19], with daily feedings at 2% of BW 2 times per day (at 07:00 A.M. and 4:00 P.M.). FSB was used as a roughage source and provided ad libitum according to each treatment. FSB left in feeders was weighed daily before new FSB allocation was provided. Daily roughage intake was calculated by subtracting feed left from feed allocated. FSB and concentrate diet were sampled daily during the last seven days and were composited by period prior to chemical analyses. All sheep were continuously kept in confined housing, with drinking water and mineral supplements available ad libitum. The current feeding trial was investigated for 4 21-day periods, allowing the animals to adapt to the pen and feed for 14 d. Every seven days during each period, feces samples were taken from each animal. Grab sampling was used to collect feces at 9:00 and 12:00 h. When feces were collected over a period of 3 h, 2 subsequent samples were combined and treated as 1 sample. Collected feed offers and feces were dried at 60 °C for 48 h, ground to pass through a 1 mm sieve in a Cyclotec laboratory mill (Tecator, Hoganas, Sweden), and kept in a humidity-controlled box until their use in chemical analysis. The apparent digestibility of DM, CP, OM, detergent fiber (NDF), acid detergent fiber (ADF), and acid detergent lignin (ADL) were calculated using the following equation: apparent digestibility = [(Amount ingested − Amount excreted)/(Amount ingested)] × 100.

### 2.3. Chemical Analysis

All samples were analyzed to determine the DM, CP, and OM according to the procedures described by AOAC [20]. Additionally, the concentrations of NDF, ADF, and ADL were analyzed according to Van Soest et al. [21]. The total number of lactic acid bacteria of the genus *Lactobacillus* were counted on MRS agar (Difco Laboratories Inc., Detroit, MI, USA) after incubating at 30 °C for 48 h in an anaerobic chamber according to the method of Kaewpila et al. [22] and then were reported as cfu/g fresh matter (FM).

At the end of each feeding period, ruminal fluid samples (approximately 100 mL) were collected at 0 and 4 h post-feeding via a stomach tube connected to a vacuum pump, and then the ruminal pH was immediately determined using a portable pH meter. Collected fluid samples were strained through four layers of cheesecloth and kept in plastic bottles. Approximately 45 mL of filtrate was dropped with 5 mL of 1 mol H_2_SO_4_, centrifuged at 16,000× *g* for 15 min, and analyzed for NH_3_–N concentration using the Kjeltech Auto 1030 Analyzer. Then, the profile of volatile fatty acids (VFAs) in the supernatants was determined using high-performance liquid chromatography. Blood samples were collected (10 mL) from the animals’ jugular veins at 0 and 4 h post-feeding on the last day of the data collection period and separated into 2 tubes. Collected blood was kept in ethylene diamine tetra acetic acid-coated tubes for the analysis of packed cell volume (PCV), plasma glucose, and plasma urea nitrogen (PUN). After blood collection, PCV was determined immediately using a microhematocrit method. For other sampling tubes, the collected blood was placed in a centrifuge separator at 1500× *g* for 10 min to obtain plasma, and the tube was frozen at –20 °C until analysis. The plasma glucose concentration was determined using an enzymatic colorimetric method and a commercial kit (Sigma Aldrich, St. Louis, MO, USA). PUN concentration was analyzed using the urea liquicolor test (HUMAN GmbH, Wiesbaden, Germany), using a spectrophotometer with a wavelength of 578 nm after a modified Berthelot reaction.

### 2.4. Statistical Analysis

The data obtained from the current research were subjected to analysis of variance for a 4 × 4 Latin square design using general linear model procedures (SAS Inst. Inc., Cary, NC, USA). The data were reported as an average from four animals with the standard error of means. Differences among groups were compared using Duncan’s multiple range test. The level of significance was *p* < 0.05, following the general model Y*_ijk_* = μ + M*_i_* + A*_j_* + P*_k_* + ε*_ijk_*, where Y*_ijk_* is the observation from sheep *j*, receiving experimental roughage *i*, in period *k*; μ is the overall mean; M*_i_* is the effect of the different experimental roughage (*i* = 1, 2, 3, 4); A*_j_* is the effect of the animal (*j* = 1, 2, 3, 4); P*_k_* is the effect of the period (*k* = 1, 2, 3, 4); and ε*_ijk_* is the residual effect.

## 3. Results

### 3.1. Chemical Content in FSB

Table 1 shows the chemical compositions, volatile fatty acids, pH, and lactic acid bacteria of the concentrate diet and FSB. The experimental animals were fed with FSB at 34.39 to 37.84% DM, prepared from the different formulations of sugarcane bagasse ensiled with or without molasses and urea. FSB contained a range of OM, CP, NDF, ADF, and ADL content (dry basis) of approximately 93.02 to 97.23%, 3.32 to 10.07%, 72.34 to 91.32%, 45.89 to 56.03%, and 8.61 to 9.98%, respectively. However, the highest CP content was found in the T4 group. Compared with the T1 group, lactic acid bacteria trended to increase in FSB with additives and showed the highest amount in the T4 group, as a result of a reduced pH.

### 3.2. Feed Intake and Nutrient Digestibility

Feed intake and apparent digestibility are presented in Table 2. There were no significant differences in the DM intake of sheep fed different FSB as roughage sources. Nevertheless, CP digestibility significantly increased (*p* < 0.05) in all FSB groups when compared with the T1 group, and the highest digestibility was found in the T4 group. On the other hand, the apparent digestibility of DM, OM, NDF, ADF, and ADL was not different (*p* > 0.05) among experimental groups.

### 3.3. Blood Parameters and Ruminal Fermentation Characteristics

Average PUN, blood glucose, and PCV did not differ among experimental treatments, except the highest PUN at 4 h was observed (*p* < 0.05) in the T4 group compared with other groups (Table 3). The proportion of butyric acid tended to be higher in all FSB groups, and the highest level was found in the T3 group (*p* < 0.05), whereas acetic acid and propionic acid did not differ among treatments (Table 4). However, there were no significant differences in ruminal pH and NH_3_-N.

## 4. Discussion

### 4.1. Improvement of Chemical Content in the FSB

The major goals of using molasses and urea are to improve the nutritional composition and utilization of sugarcane bagasse, including an increase in CP concentration and lactic acid bacteria and a rapid drop in pH. For many years, it has been widely accepted that molasses and urea can be used as silage additives to enhance the nutritive value and fermentation quality [8]. The molasses and urea treatments in the current study revealed the largest CP levels, whereas the no treatment group had somewhat lower levels of NDF, ADF, and ADL. This result was compatible with the findings of Lunsin et al. [18], who found that adding 5% molasses and 5% urea in sugarcane bagasse fermentation could enhance the CP content from 3.8% of FSB without additives to 8.1%, while the NDF and ADF were decreased after fermentation with additives. Some studies showed increases in CP content and reductions in NDF and ADF with molasses plus urea-treated sorghum silage [23] and urea-supplemented cassava top silage [24], due to conventional urea being a source of NPN (approximately 45%). Our data found that the CP content was highest in the molasses plus urea-treated bagasse, which resulted from the inclusion of urea, which increased the total N content in silage, which can then be calculated as the CP level. On the other hand, Ahmed et al. [25] noted that the fiber content was reduced in urea-treated bagasse, resulting in increased fiber degradation using an in situ method. Urea is a common alkaline additive for treating lignocellulose, which could displace lignin and increase the hydrolysis of the enzymatic activity of lignocellulosic materials [26]. These alterations of the fiber structure in the FSB treated with urea might suggest that the ammonia released from urea resulted in a more intense alkaline hydrolysis with a partial fraction of hemicellulose or lignin, resulting in a reduction of cell wall components. In addition, Ni et al. [27] reported that a lower pH also plays a key role in silage fermentation, which is related to the acid hydrolysis of plant cell walls during the ensilage process, resulting in a decreased NDF scaffold. In the present work, the use of molasses alone or in combination with urea resulted in lower pH and higher lactic acid bacteria in bagasse substrate silage than in FSB without additives. Similar results were reported by Xie et al. [28], who observed that the addition of molasses in the fermentation process can promote the growth of lactic acid bacteria, resulting in a decrease in the silage pH. Thus, the authors suggest that molasses and urea not only acted as exogenous sugar or as an N additive that reduced the pH value but also increased lactic acid bacteria during ensiling.

### 4.2. Impact on Feed Intake and Nutrients Digestibility

The intake of DM expressed as total intake, as a percentage of BW and metabolic BW, was not significantly different (*p* > 0.05). The sugarcane bagasse treated with molasses alone and in combination with urea did not affect the daily DM intake of the sheep compared with the untreated group. A similar result for DM intake was connected to the indigestible fiber content in FSB. It seems that bagasse treated with molasses and urea did not obviously alter the chemical–physical characteristics of the fibrous components in FSB. Dos Santos et al. [29] reported that the utilization of a diet rich in fiber content was limited by the physical capacity of the reticulum-rumen, resulting in a lower amount of ingested feed. According to our findings, adding molasses and urea to bagasse fermentation did not improve the digestibility of DM, OM, NDF, ADF, or ADL. Concurrently, the value of true digestibility increased in urea and Ca(OH)_2_-treated sugarcane bagasse compared with raw bagasse [30]. Lunsin et al. [18] stated that adding a mixture of 5% molasses and 5% urea to sugarcane bagasse could enhance the in vitro DM and OM digestibility. Furthermore, nutrient digestibility was higher in the sugarcane bagasse treated with *Lactobacillus*, cellulase, and molasses than in the untreated bagasse [12]. The author suggested that further experiments should be conducted to seek other silage additives that can decompose fiber fractions. Although the parameters of nutrient digestibility were not different among groups, the highest CP digestibility was observed in the bagasse treated with both molasses and urea. After consuming protein, the small intestine facilitates the absorption of amino acids and maintains the availability of amino acids to all tissues [31]. Dixon et al. [32] noted that the nutritive values of low-quality forage feed could be improved with urea supplementation, increasing the predominate source of NPN. The findings were linked to the use of urea in sheep diets, which increased rumen microbial N production and improved CP digestibility [33].

### 4.3. FSB Affected Blood Parameters and Ruminal Fermentation

In this study, the average PUN of sheep fed FSB as roughage ranged from 14.27 to 22.48 mg/dL, which resulted in the normal range (10.0 to 35.0 mg/dL) [34]. PUN, however, went elevated in the group receiving FSB with molasses and a roughage source based on urea. This information suggests that a larger ingestion of nitrogenous substances into the circulatory system is likely related to the inclusion of urea as NPN in FSB synthesis and the enhancement of CP digestibility. Blood parameters have been considered the most common measures for the assessment of nutritional alterations and the health status of the animal. Similar values of blood glucose and PCV among treatments might indicate that the sheep were healthy with a normal metabolism. Optimal conditions for cellulolytic bacteria growth and fiber digestion were positively correlated with ruminal pH in the range of 6.2 to 6.6 [13]. Decreasing ruminal pH below 6.2 reduced the digestion rate and cellulolytic activity [35,36]. The above information was consistent with our results, which suggested that a ruminal pH in all groups ranging from 6.25 to 7.43 was optimal for fiber digestion and microbial activity in the rumen. NH_3_–N is a considerable source of N in ruminal microorganisms that synthesize microbial proteins [37,38]. Some researchers have demonstrated that NH_3_–N in ruminal fluid tends to increase with increasing NPN inclusion [39,40]. Regarding our data, ruminal NH_3_–N concentrations did not differ between sheep fed FSB with and without additives. This suggests that further study should focus on the optimal level of urea that can be added in order to enhance the ruminal NH_3_-N [41,42]. The proportions of acetic acid and propionic acid did not differ among treatments. However, a positive effect of adding molasses alone and in combination with urea on butyric acid was observed. Butyrate is a significant source of energy and a significant promoter of ruminal epithelium development [43]. Ruminal butyrate can be increased to hasten the growth of calves’ forestomaches by feeding them diets rich in starch and sugar, such as those found in molasses [44]. As a result of the present addition of molasses to improve fiber digestion, significantly more butyrate is often produced when the animal digests more fiber. Additionally, the molasse addition acted as a substrate for butyrate production. Furthermore, butyrate is thought to boost the transcript abundance of proteins that mediate VFA absorption as well as growth factor secretion. The concentration of butyric acid in the ruminal fluid of sheep fed molasses-treated silage was higher than in the control groups, confirming with the findings of this study [7,39].

## 5. Conclusions

The pH and LAB population of FSB improved with the addition of molasses alone and in combination with urea, and the combination of the two increased the CP digestibility of sheep. It was suggested that sheep might utilize sugarcane bagasse fermented with 10% molasses and combined with 3% urea as an alternate roughage source without suffering deleterious effects on ruminal fermentation or the animal’s hematological status. However, more research should be conducted using a larger sample size and a longer study duration.

## Figures and Tables

**Table 1 vetsci-09-00415-t001:** Chemical compositions, volatile fatty acids, pH, and lactic acid bacteria of the concentrate diet and fermented sugarcane bagasse.

Item	Concentrate Diet	T1	T2	T3	T4
Chemical composition					
DM, %	88.04	34.39	37.84	37.61	35.27
OM, % DM	96.66	97.23	95.87	93.02	94.37
CP, % DM	13.66	3.32	4.09	5.46	10.07
NDF, % DM	39.64	91.32	84.22	75.16	72.34
ADF, % DM	24.87	56.03	54.54	47.75	45.89
ADL, % DM	6.67	9.90	9.98	9.17	8.61
pH		4.40	3.84	3.92	3.94
Lactic acid bacteria (cfu/g FM)		<10	2.4 × 10^5^	1.2 × 10^5^	9.7 × 10^5^

T1 = fermented sugarcane bagasse without additives; T2 = fermented sugarcane bagasse with 10% molasses; T3 = fermented sugarcane bagasse with 20% molasses; T4 = fermented sugarcane bagasse with 10% molasses and 3% urea; DM = dry matter; OM = organic matter; CP = crude protein; NDF = neutral detergent fiber; ADF = acid detergent fiber; ADL = acid detergent lignin; FM = fresh matter.

**Table 2 vetsci-09-00415-t002:** The total dry matter intake and apparent digestibility of Thai native sheep fed fermented sugarcane bagasse with or without additives as roughage sources.

Item	T1	T2	T3	T4	SEM	*p*-Value
Total DM intake, kg/d	0.77	0.79	0.82	0.84	5.24	0.90
DM intake, % BW	3.55	3.66	3.86	3.84	0.16	0.67
DM intake, g/kg BW^0^^.^^75^	76.28	78.69	83.74	82.28	3.38	0.62
Apparent digestibility, %						
DM	60.18	69.59	63.33	66.11	3.56	0.85
OM	70.25	71.52	72.10	73.25	2.50	0.49
CP	64.56 ^c^	69.22 ^b^	68.85 ^b^	65.56 ^a^	0.56	0.55
NDF	48.67	60.34	55.79	50.53	4.95	0.11
ADF	40.24	57.34	53.79	50.53	4.95	0.23
ADL	78.56	80.75	78.79	79.20	2.84	0.98

T1 = fermented sugarcane bagasse without additives; T2 = fermented sugarcane bagasse with 10% molasses; T3 = fermented sugarcane bagasse with 20% molasses; T4 = fermented sugarcane bagasse with 10% molasses and 3% urea; SEM = standard error of the mean; DM = dry matter; BW = body weight; BW^0^^.75^ = metabolic body weight; OM = organic matter; CP = crude protein; NDF = neutral detergent fiber; ADF = acid detergent fiber; ADL = acid detergent lignin. ^a^^–c^ Means in the same row with different letters differ (*p* < 0.05).

**Table 3 vetsci-09-00415-t003:** Plasma urea nitrogen, blood glucose, and packed cell volume of Thai native sheep fed fermented sugarcane bagasse with or without additives as roughage sources.

Item	T1	T2	T3	T4	SEM	*p*-Value
PUN, mg/dL						
0 h before feeding	15.01	14.37	15.84	18.64	4.42	0.16
4 h after feeding	16.47 ^b^	14.98 ^b^	16.21 ^b^	22.48 ^a^	0.65	0.04
Blood glucose, mg/dL						
0 h before feeding	64.25	70.00	62.00	63.75	7.81	0.85
4 h after feeding	63.50	67.50	65.50	69.00	4.47	0.87
PCV, %						
0 h before feeding	32.75	33.25	32.00	31.00	4.04	0.99
4 h after feeding	31.25	31.50	31.00	29.50	4.18	0.75

T1 = fermented sugarcane bagasse without additives; T2 = fermented sugarcane bagasse with 10% molasses; T3 = fermented sugarcane bagasse with 20% molasses; T4 = fermented sugarcane bagasse with 10% molasses and 3% urea; SEM = standard error of the means; PUN = plasma urea nitrogen; PCV = packed cell volume. ^a,b^ Means in the same row with different letters differ (*p* < 0.05).

**Table 4 vetsci-09-00415-t004:** Characteristics of ruminal fermentation in Thai native sheep fed fermented sugarcane bagasse with or without additives as roughage sources.

Item	T1	T2	T3	T4	SEM	*p*-Value
Ruminal pH						
0 h before feeding	7.43	7.35	7.45	7.27	0.21	0.75
4 h after feeding	6.55	6.25	6.26	6.45	0.22	0.07
NH_3_–N concentration, mg/dL						
0 h before feeding	4.60	4.46	4.67	5.26	2.39	0.08
4 h after feeding	13.32	11.86	14.89	16.50	1.85	0.14
Total VFA, mmol/L						
0 h before feeding	68.13	65.99	61.90	60.54	2.80	0.43
4 h after feeding	75.13	76.35	79.13	79.81	1.81	0.50
Acetic acid, %						
0 h before feeding	76.89	70.49	70.42	74.63	1.82	0.39
4 h after feeding	66.08	66.88	67.85	68.72	2.15	0.43
Propionic acid, %						
0 h before feeding	16.11	20.83	16.23	22.78	0.12	0.06
4 h after feeding	26.37	23.58	21.65	22.81	1.39	0.18
Butyric acid, %						
0 h before feeding	7.00	8.68	6.80	9.14	0.65	0.36
4 h after feeding	7.55 ^d^	9.54 ^b^	10.50 ^a^	8.47 ^c^	0.12	0.001

T1 = fermented sugarcane bagasse without additives; T2 = fermented sugarcane bagasse with 10% molasses; T3 = fermented sugarcane bagasse with 20% molasses; T4 = fermented sugarcane bagasse with 10% molasses and 3% urea; SEM = standard error of the means. ^a^^–d^ Means in the same row with different letters differ (*p* < 0.05).

## Data Availability

Not applicable.

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
