# Peer review of "Feeding Thai Native Sheep Molasses Either Alone or in Combination with Urea-Fermented Sugarcane Bagasse: The Effects on Nutrient Digestibility, Rumen Fermentation, and Hematological Parameters"

_vetsci, 2022, doi:10.3390/vetsci9080415_

Round 1

Reviewer 1 Report

Additional remarks:

The quality and the presentation of this manuscript unfortunately was unsatisfactory, I found more problematic parts in the text.

Detailed review:

Abstract:

line 37: packed cell volume!

Introduction:

lines 70-73: order of references is incorrect!

lines 98-99: the topic of aims are more expressive than title (digestibility, hematologial parameters, and ruminal fermentation)! The order of parameters in aims paragraph is correct than title!

Materials and methods

Unfortunately, this section is very unclear!

line 102: when firstly used the FSB abbreviation, please add full phrase (not enough in abstract)!

lines 119-122: theses sentences are not necessary, more focus on physiology status of animals! The “same condition” what does it mean? Nevertheless, no data available the keeping circumstance: how kept the animals? Grouped or individual?

line 122: please add the name of native sheep breed!

line 128: How measured the FSB daily intake?

line 131: crates? may be pen?

lines 131-132: when collected the faecal samples? Every day during 7-day period or the last day? How collected the faecal samples?  Individual? And when collected?

lines 140-141: how identified the lactic acid bacteria, what kind of species were found?

line 143: when collected the samples? The concentrates were given at 07:00 and 16:00, FSB ordered ad libitum!   In general, the description of sampling circumstances (methods, time) are very confusing and should be substantially improved!

line 153: If you use native tube for blood samples taken, later investigate the serum not plasma!!

Results:

Table 2-4: missing the significance column (P-value)!!

Table 2: please add fibre digestibility values!

Table 3: footnote: PCV: packed cell volume!

Table 4: please add the total amount of VFAs!

Discussion

line 230: “..NDF, ADF and ADL contents were the lowest.” but, did not find any significance level indication!

lines 268-269: not investigated the retention time, this section must be deleting!

Author Response

We appreciate your suggestion. So, we made changes in response to a specific criticism you made as well as in response to the other two reviewers. Please review the updated manuscript and revision as an attached file.

Reviewer 2 Report

Overall Comment: Whatever the merits of this paper otherwise, it suffers from a single flaw which currently disqualifies it from publication in a peer-reviewed journal - there is only 1 specimen per experimental condition. 

The author's acknowledge this severe limitation and outline the difficulty in obtaining more sheep for their trial in the methods. None-the-less the data remain non-validated until more replicated specimens of comparable weight and age are obtained and added to each experimental condition. It is simply not acceptable for any quality peer-reviewed journal to publish any data where only a single experimental repeat has been provided for all major conditions. If this major revision is met than the paper could be reconsidered for publication.

I would have several other minor comments to make if the paper addresses this major research floor and, indeed, the results of such a study may be interesting and highly publishable. 

Author Response

(The authors gave the same response as above.)

Reviewer 3 Report

Feeding Thai Native Sheep Molasses Either by Alone or in combination with Urea-Fermented Sugarcane Bagasse: Effects on Feed Intake, Rumen Fermentation, and Hematological Parameters

A straightforward work, it seems to be properly executed.

Some methodological details are needed in order to improve clarity (see file for specific comments)

My main concern as a reviewer is related to the interpretation of the effect of the molasses/urea mixture. It seems that CP digestibility as well as fiber effects could be the mathematical results of mixing the bagasse with molasses and urea (a supplement with no fiber content and 100% protein solubility). Thus, authors should provide the estimation or evidence, that there were true effects other than those obtained by dilution effect (fiber) or increased soluble protein intake (urea) (see comments in file)

Author Response

(The authors gave the same response as above.)

Round 2

Reviewer 1 Report

The quality of the materials and methods, results, and discussion sections of this manuscript were improved by authors, so I recommend this manuscript for publishing in the Veterinary Sciences journal.

Author Response

Thank very much for your kindly comment and consideration.

Reviewer 2 Report

Comments

1. Okay, so now I understand the experiment design. If I can confirm: you started with 4 sheep, each one of which was subjected to each one of the 4 treatment regimes for 21 days before they were then rotated into another one of the same four treatment regimes; such that, in total each individual sheep was subjected to each of the four treatment regimes, correct? This allows you to have n = 4 repeat animal (and samples) for each treatment. If this is the case, I apologise, I misunderstood your methods description of the experimental design. In that case you need to edit this section to make it clearer for the non-expert. I also take it that this was also implicit in the Latin-squares design you reference? This is all granted but it suggest the problem is that you need to make the experimental design section understandable for a scientist that is not familiar with it. I take it this is used in veterinary and human medical studies fairly frequently, where you are able to use subject/specimens serially through different treatment regimes. I am a biochemist who studies gut proteins and this is certainly not the case in molecular sciences. I have never seen a Latin squares design used for the very simple reason that samples are an end in themselves and cannot be re-used. Even the use of animal subjects almost always results in culling of animal for sample collection - hence the possibility of a Latin-Squares design is not possible. Please confirm that I have understood the experimental design correctly and please edit this section to make intelligible for a general biological science reader

2. I see the introduction has improved quite a bit from the initial submission. However, there are still a few things to be clarified/corrected:

·       Reference 7 is a review but you are providing research data which must have come from primary research papers. Please replace the review, or at least supplement it, with the original source of the data being provided.

·       Line 87: What is the biochemistry behind the addition of urea to silage and the supposed increased production of total (or crude protein as you term it) protein from the microbiota of the gut? Does the Gunun (ref 12) actually report this? What do you mean when you report that this study ‘improved microbial N synthesis’ are you simply referring to the activity of urease followed by glutamine synthetase to produce first ammonia and then glutamine via ammonia addition to glutamate? If so what evidence is there that this occurs in the rumen species used in the studies you are referencing. In other words, please provide references that demonstrate the required biochemistry in rumen gut to support the statements you are making here. It seems a likely point and an important one for you study yet you present it as a hypothesis i.e. ‘could’ whereas this is a well establish anabolic pathway in gut bacteria such as H.pylori in humans.

·       In addition (on the same sentence in line 87), there appear to be several addition references which give a detailed biochemical analysis of why the addition of molasses and/or urea to sugarcane bagasse increases digestibility that would be highly relevant to this sentence, e.g. doi.org/10.1080/10495398.2020.1781146, doi: 10.2527/2005.832408x, reviewed in doi: 10.3390/ani11010115. Is there any reason these and other articles are not cited and further explanation of the gut fermentation processes and nutritional value not given? There seems to be a large literature and the specific biochemistry and nutritional enhancement seems central to the background of your study. Why not spend a few additional introduction sentences explaining an providing the solid description of the gut fermentation of bagasse with molasses and urea; there seems to be ample literature for other ruminant animals than sheep, especially dairy cattle.

·       Line 91-92: if there are many examples in large ruminant animals where are the references? Preferably cite the originating ones or at least a selection of the most robust experimental papers. I found many (several dozens) by simple database searches e.g. DOI: 10.1016/j.anres.2018.11.010, DOI: 10.1007/s11250-016-1061-2

3.    3. Section 2.3: the idea of a method is so that it can be reproduced from the current paper, not by having to search through older research papers. In addition it is not unusual to find that methods cited in previous papers have changed over time and are not exactly the same as the cites source. Please detail, at least briefly, the actual methods of chemical analysis you refer here to by references 15, 16 and 17.  I see you have down this somewhat already in the new version (2). But further, on the counting of Lactobacillus, why has this been done as a normal part of the sample analysis? This is important to add, I think, as in the result (e.g. Line 181) you report the result and increases in the molasses/urea added conditions, thereby implicating it as significant. Is it simply that Lactobacillus is used as an indirect measure of general nutrient richness if the feeds? Or is it that lactate-producing bacteria measured as they play a particular role here and are an indication of an important gut lumen response to different diets i.e. pH or the ability to ferment carbon sources under anaerobic conditions? I understand that this probably has something to with the fact that the highly solube sugar content in molasses would facilitate lactic fermentation in the gut and cause a significant pH drop, but please use the methods or results to outline the logic of using this parameter explicitly.

4.       Line 123: what ‘concentrate’  this word has not been used in the manuscript before here – is this just another term for the 4 feeding regimes T1 to T4? Also see my comment on Table 1 below.

5.       The Tables require some adjustments:

·       What does the ‘Concentrate’ refer to in Table 1? I cannot find a reference to it in the results. And how is (Table) it possible that DM is only 88.04% of or ~334-37% of the composition according to the DM row. If I understand the units it is ‘Chemical composition %DM’ means the composition  as a percentage of the DM – so, again, how is it possible that in the DM row this is not always 100%? IS this mass percentage? Minor point but no-where is this explicitly written. If I have misunderstood these points it is because it is poorly explained or currently not explained at all.

·       Nice that p-values have been added to Table 2 etc. These were missing in the first draft.

·       How is it possible that you are measuring total urea content of blood (Table 3) if you are first removing the cells and analyzing only the blood plasma? If you have centrifuged away the cells (methods) then a large amount of protein (not BSA) will have been removed from blood. Do you mean, rather, that you are measuring plasma nitrogen levels i.e. PUN (Plasma Urea Nitrogen) ? Or do you have evidence in literature that sheep blood cells contain only a very small proportion of total blood (plasma + cells) urea and can , therefore, be discounted? Please explain?

·       Are these all mean percentages being report for parameters in all tables? If I have now understood the experimental design correctly (see point 1) you should have for each parameter have 4 samples, one from each animal for each of the 4 conditions T1-T4 (16 samples altogether) – please provide in the methods stats section a basic overview of the measure of center and the error term being calculated for each table value. See also the next point.

·       How is there only a single SEM error term reported for each of 4 samples/condition? Or are you reporting a SEM by combining all 16 samples for each parameter measured in Tables 2 and 3? If this is the case , this is an incorrect use of an error term. If the values reported in the tables are the means of 4 samples (1 from each animal) than you should have an error term. Where are they? This is a critical point.

·       Why have you provided a mean for Table 4 values that combining the 0 and 4 hrs value – these are different conditions and should have their own independent mean!!! I think confusion (and it is very confused) comes from your methods description of sample collection for both feces, stomach content and blood. As you have it written in the current draft it looks like you collected fecal material once per condition (i.e. ‘every seven days’) for stomach content and blood samples you state you collected samples at 0 and 4 hrs post-feeding – meaning you have taken many experimental replicates from each animal whereas for fecal matter you have collected only one per animal per experimental condition. If this not correct than you have a lot of writing to do to make this clear in your methods and to make it congruent with your table results. In any case please make this clear throughout the manuscript. Each result should state the number of animals sample/condition, the number of experimental replicate samples taken/animal/condition – both should be followed by a measure of center +/- a error term hat makes it obvious from what values these are derived and how many measurements were made.

6.       How is Digestibility measured and calculated – this is not exaplained in the methods section and makes it hard to interpret what you mean by this parameter in Table 1. I suspect it results from the analysis of the stomach content (section 2.2) but please give a detailed description of its calculation in the methods.

7.       Lines 231-233: if they have then where are the citations where scientists in this field have explicitly stated this? At least provide an example for this sort of throw-away line. If you are talking about the cumulative evidence of many studies than cite a few general up-to-date reviews (e.g. DOI: 10.3390/ani11010115)

8.       Further to point 3 above, I see you offer an explanation in section 4.1 for the role of lactic acid producing bacteria in the presence of molasses and urea – this should be outline in the methods or result when you introduce the parameters. Otherwise, the reader is left guessing until the discussion why you have measured it.

9.       Line 294: what do you mean by ‘resulted in reference value of …’ for blood urea? Is the reference simply one for the normal range of blood urea in sheep? If so just state this. In any case fix this sentence it is not understandable.

10.   Line 295: how does this suggest renal function is normal. A) you have not test renal function directly that I can see and b) urea fluctuation can be due to multiple effect beginning usually with the urea cycle in the liver, renal failure is only one and not necessarily a very common cause.

11.   Lines 196-298: what I would point out here is to ask if the small urea blood content increase you observed in the T4 compared to other conditions is really note-worthy. After all, of this is still within the normal range for sheep (see point above) than it is most certainly (as you point out).

12.   I think your discussion lacks a little focus and incorporates too much biochemical information that should have been in the introduction (see point 2 above). However, my last major question/criticism, and something that needs to be addressed, concerns your conclusion. I agree the most interesting and obvious conclusion is that the addition of both molasses and urea stimulates lactic acid producing bacteria causing a pH drop in the gut lumen. I think you should (as stated above) remove a lot of the pure background information to the introduction from the discussion and instead emphasize and focus on this point. It is not a particularly surprising finding/hypothesis and not new in animal husbandry (although looks as you have claimed in the abstract) to be novel for sheep. If you have not done so I would also point out that the reason protein digestibility is improved is for the simple reason that lactic acid increase, and subsequent gut acidification, will promote acid hydrolysis of peptide bonds. However, what is not well explained or discussed is why this would necessarily lead to increase of amino acid uptake into the blood (portal vein). This is suggested by the increased NH3-H content in blood (almost certainly due to increase arginine in the liver.  What you are missing is an explanation and discussion of how the acidification of the gut (effect of molasses/urea) would effect the uptake (absorption) of protein hydrolysis products (small peptides and amino acids) in the gut. You need to add a small discussion (a few sentences) that makes the link between improved gut digestibility and increased nitrogen blood levels. Although there maybe little primary research on protein absorption in sheep or ruminant guts, there is a substantial literature on humans and mammals in general (rodents, rabbits etc) which maybe of useful reference to you. See doi: 10.1113/EP085029 and doi: 10.1002/cphy.c170041, as the most comprehensive updated reviews on mammalian amino acid and peptide absorption in the gut. Both should be cited in your discussion with reference, briefly, to the various mechanisms for how protein hydrolysis products would enter the portal vein of sheep. This is especially relevant in light of the pH-dependence of multiple transports in the gut epithelium that are responsible for this absorption process.

13.   Why is butyric acid synthesis (fermentation) not really suppressed in conditions where molasses is added (T2 and T3)? According to the basic effect of sudden pH-reduction in the gut (molasses driven lactic acid fermentation) this would retard butyrate fermentation (see Fink-Gremmels, J. Mycotoxins in forages. In The Mycotoxin Blue Book; Duarte, D.E., Ed.; Nottingham University Press: Nottingham, UK, 2005; pp. 249–268. For the background knowledge). Why do you not see this with the addition of molasses alone (T2 and T3). Why do think it is reversed in the T4 condition (addition of Urea)?

Author Response

We appreciate your suggestion. We did our best to edit and revise the text according to your recommendations. Please see our point-by-point response as an attached file. Thank you very much for your kind consideration. 
